# Explainability-Informed Feature Selection and Performance Prediction for Nonintrusive Load Monitoring [note 1]

**DOI:** 10.3390/s23104845

**Published:** 2023-05-17

**Authors:** Rachel Stephen Mollel, Lina Stankovic, Vladimir Stankovic

**Affiliations:** Department of Electronic and Electrical Engineering, University of Strathclyde, Glasgow G1 1XW, UK; lina.stankovic@strath.ac.uk (L.S.); vladimir.stankovic@strath.ac.uk (V.S.)

**Keywords:** NILM, decision tree, explainability, multiclassification

## Abstract

With the massive, worldwide, smart metering roll-out, both energy suppliers and users are starting to tap into the potential of higher resolution energy readings for accurate billing, improved demand response, improved tariffs better tuned to users and the grid, and empowering end-users to know how much their individual appliances contribute to their electricity bills via nonintrusive load monitoring (NILM). A number of NILM approaches, based on machine learning (ML), have been proposed over the years, focusing on improving the NILM model performance. However, the trustworthiness of the NILM model itself has hardly been addressed. It is important to explain the underlying model and its reasoning to understand why the model underperforms in order to satisfy user curiosity and to enable model improvement. This can be done by leveraging naturally interpretable or explainable models as well as explainability tools. This paper adopts a naturally interpretable decision tree (DT)-based approach for a NILM multiclass classifier. Furthermore, this paper leverages explainability tools to determine local and global feature importance, and design a methodology that informs feature selection for each appliance class, which can determine how well a trained model will predict an appliance on any unseen test data, minimising testing time on target datasets. We explain how one or more appliances can negatively impact classification of other appliances and predict appliance and model performance of the REFIT-data trained models on unseen data of the same house and on unseen houses on the UK-DALE dataset. Experimental results confirm that models trained with the explainability-informed local feature importance can improve toaster classification performance from 65% to 80%. Additionally, instead of one five-classifier approach incorporating all five appliances, a three-classifier approach comprising a kettle, microwave, and dishwasher and a two-classifier comprising a toaster and washing machine improves classification performance for the dishwasher from 72% to 94% and the washing machine from 56% to 80%.

## 1. Introduction

The large-scale, worldwide, smart meter roll-out together with in-home displays, which can provide real-time information on aggregate energy consumption at the building level, is enabling disaggregation of energy consumption down to individual appliances or loads at any point in time without additional metering. One of the benefits of load disaggregation is to help consumers to better manage their energy consumption and bills. Since load disaggregation or the nonintrusive load monitoring (NILM) technique was first proposed, several approaches based on machine learning have been developed focusing primarily on improving disaggregation accuracy using ever more complex machine learning approaches [1,2] that are mostly treated as “black box” models. However, the challenges, as summarised in [3] to realizing the trustworthiness of of NILM, and therefore its wider adoption, remain its generalisability across different load profiles, models enabling continuous learning and embedding user feedback, explaining NILM outcomes, fair performance evaluation, and developing models that are privacy-preserving. Furthermore, the European Commission has recently published seven principles of trustworthy AI [4], which include human agency and oversight; technical robustness and safety; privacy and data governance; transparency, diversity, nondiscrimination, and fairness; societal and environmental well-being; and accountability. This paper aims to address some of these principles of a trustworthy AI-based NILM system design as follows: (1) developing a human-in-the-loop approach which enables human intervention during the design cycle of the NILM system and monitoring the system’s operation via output explainability plots; (2) designing NILM systems for high accuracy but also highlighting how likely errors are for occasionally inaccurate predictions; (3) attaining reliability of design by ensuring that the NILM design works for a range of inputs and situations by demonstrating performance on different houses with different appliances; (4) including traceability to enable transparency by leveraging public datasets, where data gathering, labelling, and performance with different algorithms are well documented, (5) ensuring explainability for transparency by providing explanations of the NILM system’s decision-making process; (6) providing communication for transparency by clearly identifying the level of accuracy and limitations; (7) using a low-complexity methodology to ensure implementation is environmentally friendly without resorting to large data centres since the system can run locally; and (8) addressing the UN Sustainable Development Goal 7 by enabling responsible consumption of energy and an affordable and modern energy service.

Specifically, this paper proposes a simple and reliable methodology to codesign an NILM system with the target building owner in which an explainable supervised decision-tree-based multiclass classifier is built that requires few samples of electrical measurements from the smart meter data labelled by the owner. The proposed multiclass classifier then leverages upon explainability tools to determine the best features that would result in the best prediction outcome for the target appliances of the building (informed by the occupants), resulting in a multiclass classifier design with feature selection to guarantee the best performance for each target appliance instead of the best performance on average. This simple approach ensures that the aggregate measurements never leave the house and that disaggregation is carried out locally in the house.

An NILM model is said to be explainable or interpretable if the reason behind its prediction can be explained by the end-user. The importance of explainability is described in [5,6] to be the following: (a) to facilitate learning and satisfy curiosity as to why certain decisions have been made by the model to build trust, (b) to learn which are the important features that contribute significantly to the outcome and which are not for the purpose of parameter tuning, and (c) to debug the model in case of errors. Explanations for the machine learning model can either be model-specific or model-agnostic. The latter can be applied to any model, while the former only works for one model type. Model-agnostic methods can be further classified into local and global methods. Global methods focus on how features affect the prediction on average and local methods aim to explain individual predictions [6].

This paper builds upon our conference paper [7], where model-agnostic explainability tools, in the form of global partial dependence (PD) plots and local individual conditional expectation (ICE) plots were used to inform feature selection for the design of a DT-based NILM five-class classifier. Perfect event detection was assumed since the focus was on which features impacted the multiclassifier.

The contributions of this paper are as follows:Using a methodology that enables the codesign with the building owner or householder for a scalable, trustworthy, and privacy-preserving NILM, comprising event detection, feature generation and selection, and DT-based multiclass classification using only smart meter readings as input;Leveraging post hoc model-agnostic global and individual explainability of the aforementioned multiclass classifiers’ models to inform feature selection for each appliance and subsequent design of the DT multiclass classifier;Explaining how one of more appliances within a multiclass classifier can negatively impact classification of other appliances leading to different multiclassifier models for different sets of appliances;Predicting the performance of the trained DT multiclass classifiers for each target appliance via PD and ICE plots, validated during testing by standard; classification performance metrics on unseen data from the same house;Predicting the performance of a trained DT multiclass classifier on unseen houses from the same dataset and from another dataset to explore generalisability and transferability.

The paper is organised as follows. Section 2 describes the related work on the NILM multiclass classification problem and explainability for NILM. The proposed methodology for event detection and explainability-informed multiclass classifier design are described in Section 3, which is followed by a description of the experimental setup and event detection performance in Section 4. Performance prediction and explainability-informed multiclass classifier design are described by example, using the REFIT and UK-DALE datasets, for the target house and to other houses in Section 5 and Section 6, respectively. Finally, the key findings are discussed in Section 7.

## 2. Background and Related Work

As stated in Section 1, multiple approaches for tackling the NILM problem leveraging the availability of smart meter roll-out measurements (active power at resolution of 1 s to 60 min) have been reported in the literature over the past decade; see [1,2] for a detailed review. Of these, as expected, supervised NILM approaches tend to outperform unsupervised approaches, with variable data quantity requirements for training. NILM has been tackled as a classification problem, through detecting when particular appliances are switched on or as a regression problem, through estimating the energy consumption of individual appliances. NILM classifiers are useful to understanding routines in households and tend to underpin demand response measures, smart home automation, and activities of daily living. Therefore, it is important for the base NILM classifier to be robust and explainable to enable trust for the wider adoption for different applications.

Popular methods that have been used for explainability are PD/ICE plots [8] that show the relationship between features and predicted outcome, feature importance that gives the score for all input features based on how useful they are for the prediction outcome, LIME [9] that focuses on explaining the individual predictions by monitoring model behaviour when the input feature changes, and Shapley value that measures the impact of an input feature to the predicted outcome, taking into account the interaction with other input features [10].

In this section, we first provide a short background on NILM, then describe the DT-based approaches used as benchmarks for NILM, and finally review the early attempts at explainability for NILM.

### 2.1. NILM

Different approaches have been proposed over the years for NILM to operate on low-rate smart meter readings (less than 1 Hz), using various signal processing, machine learning, and deep learning techniques, and which evaluate a range of household appliances. Unsupervised approaches that do not require labelled training sets include the hidden Markov model (HMM), unsupervised graph signal processing (UGSP), and dynamic time warping (DTW); the latter two have been shown to outperform HMM methods. HMM approaches are inefficient when the number of disaggregated appliances increases and they have high computational complexity [1], which is not the case for UGSP-based methods [1,11]. Popular supervised NILM approaches include decision tree (DT) [12], boosting-based ensemble algorithms [13], and deep learning (DL)-based approaches (see [2] for a review), all of which require well-labelled datasets for training the models. Supervised ensemble and DL approaches are capable of obtaining good generalisability and transferability to houses never seen before, but larger training datasets are required and are often complex [1,2]. Semisupervised approaches, such as supervised graph signal processing (SGSP) [14] have been proposed to keep training sets small.

NILM algorithms are either model-based (e.g., DL-based approaches) or event based (e.g., DT, GSP). The former does not rely on event detection, while the latter does. The types of features that are generated for input to NILM depend on the sampling frequency and include raw power measurements, active power, change in power, power factor, and reactive power for low-frequency NILM; and harmonics, wavelet coefficients, etc. for high frequency NILM [15]. Features are categorised into steady state (e.g., power, power factor, reactive power) and transient (e.g., transient root mean square current, transient duration). The steady-state features are straightforward and easy to generate but can result in misclassification for appliances with a similar power rating [16]. Transient features can improve classification performance, but they require a complex hardware setup, high sampling speed, and heavy computation [15,16]. Selecting a set of optimal features is constrained by the data availability (open-source datasets and the available features are summarised in [17]). Low-rate NILM approaches, which are suited for nationwide-deployed smart meter data, can only use steady-state features, typically active power measurements [18].

### 2.2. DT-Based Multiclass Classifiers for Low-Rate NILM

DT models form a hierarchical structure with nodes and branches that can easily be followed through (from a parent node to leaf nodes) to understand how the outcome is generated. For classification, the metrics often used for best splitting decisions in DT are Gini impurity and information gain [19]. Gini impurity measures the probability of a certain class being incorrectly classified, while the information gain is a metric that chooses the split based on how much the entropy has been improved. DT is therefore interpretable by design, in the sense that it is possible to design a tree in way that decision outcomes can be mathematically explained and predicted. Logistic regression, rule fit, and Naïve Bayes are also inherently interpretable on a modular level for the classification task [6]. Of these, DT is the better-performing approach and most suited for the multiclassification task. However, as the tree becomes more complex with numerous decision splits, the dependence of a predicted outcome on each feature is not easily seen. In other words, it is often difficult for a human to infer how the outcomes have been generated. Therefore, additional explainability methods are needed to shed light on the most important features that steer the model towards a particular decision.

In [12], DT, for low-frequency active power measurements, was used for the classification of multiple common household appliances for the classification of a pancake maker, washing machine, hair dryer, oven, television, kettle, boiler, microwave, toaster, washer–dryer, and fridge. Active power changes during appliance state transitions were considered for identifying appliances while ignoring power fluctuations within each state. Active power changes occur at the switching on/off of an appliances or when the appliance changes states. The DT-based method showed good performance on the house it was trained on but poor performance for the toaster and dishwasher in unseen houses. Only a small dataset (one week of data) of aggregate active power for training was used and thus required the least storage and computational resources compared to the unsupervised methods, HMM and DTW. A DT-based classifier implementation of [12] that classified five appliance classes, namely the fridge, kettle, toaster, microwave, and dishwasher, was evaluated in [14] in comparison with their proposed SGSP approach, demonstrating good classification performance for all 5 appliances on the REFIT and REDD datasets with postprocessing of DT and SGSP outputs using regularisation. Furthermore, the authors of [20] evaluated DT with the change in power as the feature for classifying the dishwasher, washing machine, tumble dryer, and washer-dryer in the REFIT and REDD datasets, with various preprocessing approaches in relation to supervised KNN and unsupervised DBSCAN approaches, and concluded that DT had the best classification and disaggregation performance for all appliances of interest, comparable to state-of-the-art algorithms, and needing very little training data.

None of the previous DT multiclass classifiers reviewed above considered feature selection or explainability.

### 2.3. Explainability for NILM

As the research field on NILM grows, there has been a need to provide explainability of learning models to foster trust in end users [3]. In [5], heatmaps are demonstrated as a model-agnostic way to visually interpret time-series results for DL models to explain NILM outputs. The paper used a sequence2point network to illustrate explainability. To explain how the model makes decisions, part of raw energy data input is occluded, and the sliding of the occlusion window across data is performed. For each window position, the model’s singular point output is estimated and the heatmap is generated. The heatmap shows what the model considers to be the most impactful features. It was also shown that the metrics, commonly used for evaluating the performance of deep learning approaches, are not truly explainable since they are not necessarily intuitive. Heatmaps may be difficult to explain to the end-user, who has little to no domain knowledge.

Another work that used a heatmap is presented in [21]. The paper describes using NILM convolutional neural network (CNN) classifiers and two explainable artificial intelligence (XAI) techniques, occlusion sensitivity and gradient class activation mapping, to provide simple feedback to the consumer-user. A CNN model for NILM is used to estimate the activation state of each appliance in the system. Then, a XAI technique uses this model and explains and justifies the prediction of the model. The explanation from the XAI technique is presented as a heatmap such that the value of each variable in it indicates how important each input feature is to the CNN output.

LIME was implemented with a neural network (NN) model in [22], using household transportation energy (HTE) consumption (used as a model output/label for the prediction model). HTE depends on household trip generation, the travel mode involved, fuel type, and trip distance, which is essential in decision-making among urban planners and building and transportation engineers. LIME provides local explainability and has the advantage that it can highlight which features are considered for certain outputs and how much more important one feature is than another for a specific prediction [22]. First, LIME is used to explain the local inference mechanisms on individual (household) predictions. Second, SP-LIME (submodular pick-LIME), a method that picks a set of illustrative instances with explanations, is used to address the problem of trust.

As previously explained, DT is interpretable by design. However, for complex tree models such as those used for NILM, it is very difficult for a human to interpret the output of the model based on the tree structure. Furthermore, feature importance cannot be assessed easily. Hence, explainability approaches are needed to interpret the output of the model. In our preliminary work [7], PDP, ICE plots, and feature importance were used with DT to inform the model performance. PD plots show how the features affect the predicted outcome of the global model. PD plots and feature importance provide a global explanation of a model, i.e., they quantify the importance of each feature. Feature importance is calculated as the number of samples of a feature that will reach the leaf node (predicted outcome) divided by the total number of samples of that particular feature. The higher the value is, the more important the feature. ICE plots, similarly to PD plots, visualize the relationship between features and the predicted outcome but on an instance basis. With ICE, it is possible to explore the relationship between a particular instance or group of instances of a feature with the predicted outcome, i.e., it is easy to observe each instance in a feature and its relationship to the predicted outcome. Knowing the effect of each feature on a model’s prediction is essential in understanding how the model will behave with new data. It has been observed that although the overall model considers specific features as more important than others, local explainability is critical to explaining false positives. In addition to the aforementioned explainability techniques, we also make use of box plots [23,24,25] to explain the uncertainty of each generated feature. This is due to noise from unknown appliances and in turn reflects upon the local feature importance in the ICE plots and impacts the complexity of the learnt splits in the DT model.

## 3. Explainability-Informed NILM Multilabel Classification

Figure 1 describes the overall NILM workflow. As discussed in Section 2, the input to the classifier is the aggregate smart meter power readings. Automated edge detection is performed on the aggregate power measurements of the training dataset to the generate features used. These features are then fed to the multiclassifier for training. Explainability is then performed on the trained model, providing the input to new models developed via feature selection. Next, we describe each of the three steps, one by one.

### 3.1. Automated Event Detection

Let $P_{t}$ be the aggregate power measurement at sampling instant *t*, comprising the sum of power consumption of known individual appliances, $P_{t}^{i},i=1,\ldots ,M$, and measurement noise together with noise due to unknown appliances $e_{t}$. That is, $P_{t}=\sum_{i=1}^{M}P_{t}^{i}+e_{t}$. The difference between two adjacent active power samples is denoted by $\Delta P_{t}=\left|P_{t}-P_{t-1}\right|,t>0$.

DT is an event-based method, requiring isolated events when appliances are running. The appliance on/off state is detected using an appliance-specific thresholds $T^{i}$. The threshold set of values is denoted as T = $\left\{T_{1},\ldots ,T_{M}\right\}$. Let $min\left(T\right)=min_{i}$T, i.e., the smallest threshold value among all *M* appliances. First, all Δ*P* values below min(T) are removed. Next, for each appliance *i*. the algorithm identifies the first potential rising edge of the appliance *i* via the following: (1)$$P_{t}-P_{t-1}\geq T^{i}.$$

Let $t_{p}^{i}$ be the identified position (i.e., time stamp) of the rising edge. The corresponding falling edge is found by searching for the first $t,t_{p}^{i}<t<t_{stop}$ that satisfies Equation (Equation 1) after the left-hand side is multiplied by −1. This time instance is recorded as $t_{n}^{i}$. If no such *t* exists, the rising edge is discarded and the algorithm starts searching for the next rising edge. $t_{stop}$ denotes a time window parameter, which is set to be less than 5 min and greater than 30 min for shorter and longer operating appliances, respectively.

For each appliance, the algorithm will output the following features: $\Delta P_{t_{p}}$ denoted by EDGE_P (in Watts), $\Delta P_{t_{n}}$ value when the appliance goes OFF denoted by EDGE_N (in Watts), and duration, named DURATION$=t_{n}^{i}-t_{p}^{i}$.

### 3.2. DT-Based Multiclass Classification

DT classification is used to predict categorical values, which in our case, are the appliances used in the household. The aim of the multiclass classification is to assign to each aforementioned detected event an appliance label. Being a supervised algorithm, the DT classifier requires labelled data during training. To focus on high-consuming household appliances that are typically present in all homes as well as for easy comparison and benchmarking, first, labelling is performed into five classes, namely kettle, microwave, toaster, dishwasher, and washing machine. Second, the generated features (EDGE_P, EDGE_N and DURATION) are used as input features together with appliance labels during training. After training, the model is exported for prediction on unseen data without labels.

The tree model is a hierarchical structure consisting of nodes and branches. At the nodes, the decision based on the input feature is made, and the leaf node is the last node representing the outcome of the decision. During the training phase, the decision tree splits the nodes from the root node to the leaf node based on training data (extracted labelled features). Note that the learning algorithm will attempt all possible splits with all available input features to find the best split. The splitting process will continue iteratively until the decision tree separates the whole training set as observed in Figure 2 depicting our five-classifier. The interpretability decreases as the depth of the tree increases. From the tree, we observe that the tree is pretty simple to understand from the root node to the leaf node “Kettle”, “Microwave”, “Toaster”, and “Washing Machine” (when DURATION is greater than 770 s) showing which feature contributes to the which outcome at the leaf node. However, the splits after the node with feature DURATION greater than 550 s becomes more complex. It then becomes difficult to assess the effect of each feature on the predicted outcome. Hence, the importance of additional explainability tools.

In order to explore the influence of appliances with similar features, the following multiclass classifier models are built and trained:Multiclass classifier for classifying the five labels of kettle (K), microwave (M), toaster (T), dishwasher (DW), and washing machine (WM), referred to as DT (K-M-T-WM-DW);Multiclass classifier for classifying the three labels of kettle, microwave, and dishwasher, referred to as DT (K-M-DW);Multiclass classifier for classifying the two labels of toaster and washing machine, referred to as DT (T-WM).

In order to predict certain outcomes during testing, after the features of the testing samples are generated (namely, EDGE_P, EDGE_N, and DURATION), the generated feature proceeds through the trained tree from the root node to the leaf node. The feature is observed at the node, and depending on the condition at this node, it is assigned either to the left or right tree branch. The process continues until the feature reaches the leaf, and the corresponding label assigned to this feature.

### 3.3. Post Hoc Explainability: Predicted and Actual Outcomes

Once the multiclass classifier model is trained, first, we plot the feature importance graph. Feature importance is a graph that shows which features the model considered strongly in the prediction (the feature considered more important will have the highest score). However, global feature importance does not show the effect of individual features on individual predicted outcomes, which is important in explaining false positives. This is observed from PD and ICE plots.

PD and ICE plots are usually presented in a single plot. A PD plot is presented as a single curve showing how a feature generally influences the model outcome. This is similar to feature importance, but with a PD plot, we can show graphically how much an individual feature influences each predicted outcome because it enables for the plotting of each feature for each outcome. The y-axis of the PD and ICE plot shows the score of the predicted outcome (between 0 and 1) with respect to the feature and with respect to instances of each feature in the ICE plots. The smaller the PD curve is, the less effect a feature has on the model prediction. An ICE plot provides an explanation at the instance level; hence, it indicates how the individual instances of a feature are distributed. The instances that appear to have almost one “score” in the ICE plots have a very high degree of impact on the predicted outcome. This is shown by example in Section 5.

PD and ICE plots are also useful for explaining why the actual outcomes differ from the predicted outcomes. This is carried out by comparing the PD and ICE plots of the predicted outcome with the PD and ICE plots of the actual outcomes. The differences explain the limitations of the trained model when it experiences test data that differ from the trained data, e.g., from another house in the same dataset or from another house in another dataset. This is shown by example in Section 6.

## 4. Experimental Setup

### 4.1. Dataset

The five appliances of interest, namely the kettle, microwave, toaster, dishwasher, and washing Machine, are present in both Houses 2 and 6 of the REFIT dataset [26] and House 1 of the UK-DALE dataset [27]. The sampling rate for REFIT houses is 1/8 Hz and UK-DALE 1/6 Hz. For the sake of data transparency, these datasets, appliances, and chosen houses are the most documented since they are the ones mostly used in the literature to demonstrate performance and hence useful to obtain a true indication of performance.

Training is carried out on a balance set of edge-pairs (55 edge-pairs per each appliance) selected randomly from the available data of House 2 (except for the test period) and tested on the entire unseen months of October, November, and December 2014 of House 2.For generalisability and transferability experiments, the aforementioned trained model from REFIT House 2 is used for testing on unseen REFIT House 6 (the month of October 2014) and UK-DALE House 1 (January 2016 to April 2016).

### 4.2. Evaluation Metrics

Training and ten-fold cross-validation were performed on our classifier models, and the results are presented in confusion matrices of Tables in Section 5.2, Section 5.3, Section 6.1 and Section 6.2. During evaluation, we consider the influence of unknown appliances contributing to noise from unlabelled appliances, $e_{t}$. Each predicted event is compared with submetred data as ground truth data and labelled as either true positive, false positive, or a miss. This, in turn, is used to calculate the performance metrics. For performance evaluation, the following metrics were used: precision (*PR*), recall (*RE*), and *F-score*.

These metrics are expressed as Equations (Equation 2)–(Equation 4).
(2)$$PR=\frac{TP}{TP+FP}$$
(3)$$RE=\frac{TP}{TP+FN}$$
(4)$$F\text{-}Score=\frac{2\times PR\times RE}{PR+RE}$$
where, for Appliance *i*, a true positive (*TP*) indicates that the classifier has correctly classified Appliance *i* as Appliance *i*, false positive (*FP*) indicates that the classifier has incorrectly classified another appliance as Appliance *i*, false negative (*FN*) indicates that the classifier has incorrectly classified appliance *i* as another appliance. Precision (*PR*) is the ratio of correctly classified Appliance *i* samples to the total number of samples predicted as “true”, recall (*RE*) is the ratio of correctly classified Appliance *i* samples to the total number of Appliance *i* activation, and *F-score* is defined as the harmonic mean of the model’s *PR* and *RE*.

### 4.3. Detection Performance

For the purposes of transparency, we also evaluated the performance of the edge detection algorithm since this influences the overall performance. The edge detection algorithm detects the appliance ON events by monitoring the change in active power in relation to the appliance-specific thresholds. The appliance-specific thresholds $T^{i}$ estimated from the average appliance submetring measurements for House 2 REFIT dataset were as follows: (a) 2000 W for kettle, (b) 1900 W for dishwasher and washing machine, (c) 1000 W for microwave, and (d) 700 W for the toaster. The same thresholds were used to evaluate the generalisability and transferability to other unseen houses.

Table 1 shows the edge detection performance in terms of the number of events detected (i.e., number of EDGE_P and EDGE_N pairs detected) that correspond to the five appliances of interest and events that were due to ‘Other’ unlabelled appliances in relation to the total number of ground-truth events corresponding to the five appliances of interest. The latter events could cause FPs for our five appliances of interest and therefore affect the precision and F-score of the classification performance. When perfect edge detection is assumed, events caused by “Other” unknown appliances in the dataset are not passed on to the classifier.

Low-rate NILM is usually challenging to disaggregate due to the presence of unknown appliances. This can be calculated using the noisiness of the dataset metric of [28] as per Equation (Equation 5), as shown in Table 1 for each of our test houses. The higher the percentage is, the higher the noise.
(5)$$\%-NM^{\left(T\right)}=\frac{\sum_{t=1}^{T}\left|y_{t}-\sum_{m=1}^{M}y_{t}^{\left(m\right)}\right|}{\sum_{t=1}^{T}y_{t}}$$
where $y_{t}$ is the aggregate load measurement at time *t*, $y_{t}^{\left(m\right)}$ is the ground-truth power measurement for each appliance *m* to be disaggregated, and *T* is the testing period as specified in Section 4.1. *M* = 5 where the only *m* of interest are the kettle, toaster, washing machine, dishwasher, and Microwave. All the other loads are considered to be noise. From Table 1, we can observe that the noise metric for REFIT House 2 and UK-DALE House 1 are about the same, resulting in a similar percentage of detected events with similar active power from unknown appliances. However, the noise measure in REFIT House 6 is significantly higher, resulting in a greater number of events from unknown appliances with similar power being detected. This likely affects the classification performance, with a worse classification performance expected for REFIT House 6 compared to the other two.

### 4.4. Execution Time

Computational complexity and storage requirements have been a challenge to most NILM applications. While edge computing has been proposed for deep learning-based NILM showing reduction in model complexity and storage requirements along with small performance degradation [29], DT, being of relatively lower training and testing complexity, is suitable for both hardware and software implementation with parallelism without performance degradation [30].

The execution time of our proposed DT multiclass classifiers is summarized in Table 2. The experiments were performed on an Intel(R) Core(TM) m7-6Y75 CPU 1.2 GHz machine running Windows 10. The models were all implemented using MATLAB 2023a [31] classification learner with an optimizable tree that can optimize hyperparameters, i.e., the maximum number of splits and splitting criterion such as the Gini index and twoing criterion.

It can be observed that the most complex model requires only 40 s for training with 2 months’ worth of data, while testing can be performed in real time, requiring around 1.52 milliseconds for one hour of data or only 4.24 ×$10^{-7}$ s per sample. The table also shows minor variations of the execution time due to training sizes, model complexities, and class composition.

## 5. Performance Prediction via Explainability Tools

In this section, we first explore the global feature importance of the model, then use the PD and ICE plots to establish the predicted outcomes of the DT multiclass classifiers described in Section 3.2 per appliance and per feature. These are then validated against the observed confusion matrices and performance metrics.

### 5.1. Global Feature Importance

Figure 3 shows feature importance scores for all three models described in the Section 3.2.

Figure 3 shows that DURATION is the most important feature in a 5-appliance classifier model on average across all five appliance classes. EDGE_P and DURATION have the same importance in a three-appliance classifier model on average across all 3 appliances. For a two-appliance classifier model, EDGE_P is the important feature on average across the two appliances. Note that the feature importance captures the contribution of each feature to the model after all previous features have been added. For example, in Figure 3, the DT (T-WM) model, EDGE_N and DURATION do not provide any improvement once EDGE_P is introduced.

### 5.2. DT (K-M-T-WM-DW) Multiclassifier Model

The PD and ICE plots for this five-class classifier were described in [7] and are summarised below.

Figure 4, Figure 5 and Figure 6 show the obtained PD and ICE plots. Figure 3 shows that, in a five-appliance classifier, DURATION is the most important feature with a score of 0.02 while EDGE_P and EDGE_N have similar importance with a score of 0.016 and 0.017, respectively. However, whilst this holds true for the dishwasher and washing machine, EDGE_P has an equally strong influence on kettle performance as does DURATION (a 0.6 PD score in both Figure 4a and Figure 6a). EDGE_P has a significantly stronger influence on microwave prediction compared to DURATION (0.78 vs. 0.2 PD score as observed in Figure 4b and Figure 6b, respectively). The ICE plot of Figure 4a shows that the individual instance scores of the kettle are very well clustered and rarely mixed. Indeed, all EDGE_P below 1800 W have a score of 0, and all values above 2700 W have a score of 1 (highlighted in green). The only issue appears with a single EDGE_P instance of around 2400 W that is mixed with the 0-score cluster of instance points between 2000 W and 2600 W (highlighted in yellow), where the kettle is likely to be confused as either the washing Machine or dishwasher, which is also shown in Figure 7a as an outlier. This single instance causes the Kettle’s PD plot in Figure 4a to rise by only 60%. DURATION, as a feature as shown in Figure 6a, also influences kettle performance with a similar score as EDGE_P. Therefore, we can expect excellent performance for the kettle, with EDGE_P and DURATION as features.

The microwave and toaster have the lowest and similar power consumption and a similar operation duration. Therefore, they tend to be confused for each in the learning of split decisions, as observed by multiple instances in EDGE_P around the same wattage in Figure 4b,c (multiple overlapping score 0 and 1 clusters highlighted in yellow). This can be observed from Figure 7a as well, although the microwave and toaster box plots do not overlap (indicating that the individual instances of EDGE_P for the microwave and toaster are different) but the outliers do. The microwave has high prediction score because, between 1240 W and 1530 W (where we observe no more outliers in toaster that may affect the microwave between these values as shown in Figure 7a), the cluster is well separated (highlighted in green). As shown in Figure 5b,c, EDGE_N is a better feature for distinguishing between the microwave and the toaster. EDGE_N instances less than −1180 W are considered strongly for the microwave, and more than −1180 W are considered strongly for the toaster (highlighted in green). From Figure 7b, we can observe only about four outliers in the toaster spread between −1700 W and −1344 W that are overlapped with the microwave box plot compared to what was observed with EDGE_P as a feature. DURATION has very little influence on either the microwave or toaster performance, as shown by the relatively low PD scores in Figure 6b,c as well as by Figure 7c, where the box plots and outliers overlap. Therefore, we can expect excellent performance for microwave with EDGE_N as a feature. However, EDGE_N, with a PD score of 0.39 for both the toaster and microwave (as seen in Figure 5b,c), may help distinguish these two appliances.

The dishwasher and washing machine tend to cause confusion in the learning of split points, as observed via their ICE plots. EDGE_P shows very low confidence in the prediction outcome of washing machine and dishwasher as seen in Figure 4d,e and observable by EDGE_P instances that are distributed around the same wattage, as seen in Figure 7a. With EDGE_N, the model leans towards the dishwasher between −2280 W and −2567 W, and between −2215 W and −2038 W, but lean towards a washing machine prediction with EDGE_N instances between −2281 W and −2216 W, and instance at −1937 W. Although there is a rise of their PD plots in Figure 5d,e at those wattages, the PD plot scores are very low, indicating low confidence in prediction. Confusion can also be observed via the overlapping box plots and outliers between the washing machine and dishwasher data in Figure 7b. With DURATION as the feature, instances are not as spread, as shown in Figure 7c for the dishwasher as in washing machine, and no outliers appear that can cause confusion; although their respective box plots overlap, it can be clearly seen from Figure 6d,e, that some DURATION instances have more impact on one appliance than they do on the other. For prediction “Dishwasher”, most of the DURATION values between 570 s and 780 s have a high impact. For prediction “Washing Machine”, values between 270 and 540 s, and higher than 780 s, have a higher impact, as supported by their PD plots in Figure 6d,e, which shows a rise of about 50%. Hence, we expect good performance for the dishwasher and washing machine with DURATION as a feature since it helps distinguish between these two appliances.

In summary, kettle is expected to work best with EDGE_P and DURATION as features, microwave with EDGE_P as a feature, washing machine and dishwasher with DURATION as a feature, and to a lesser extent EDGE_N to help distinguish between toaster and microwave. To confirm this, we performed testing on an unseen portion of the same House 2 dataset with three 5-class classifier models embedding feature selection. Table 3 shows the resulting F-scores when the models were first built with all three features, second with EDGE_P and DURATION as features, and third with EDGE_N and DURATION as features, with their confusion matrices presented in Table 4. Note that, unlike the performance results reported in [7], where we assumed perfect edge detection and therefore did not consider the influence on events due to unknown/unlabelled appliances, here we take into consideration these unknown appliances, which we refer to as “Other”.

We expect the best overall performance for the second model since for all appliances, except for toaster, EDGE_P and DURATION are considered to be strong influencing features. This is indeed supported by the F-scores of the five-class classifier models, with EDGE_P and DURATION selected as features. The effect of EDGE_N on the toaster can be observed in the confusion matrix of Table 4c where the toaster is less confused with microwave. However, this is not reflected in the equivalent F-score in Table 3 because of the 16 instances of Other, i.e., unknown appliances with similar EDGE_N to the toaster, that increase the number of false positives for toaster and reduce its precision and thus F-score. This effect of "Others" has less effect on the five-class classifier with EDGE_P and DURATION as selected features.

Furthermore, we can observe from the confusion matrix in Table 4b with EDGE_N and DURATION selected as features that the dishwasher and washing machine classes are confused with each other since there is not a clear distinguishing feature between these two appliances, as shown in the PD and ICE plots. Therefore, it makes sense to design a multiclass classifier where the DT does not need to distinguish dishwasher and washing machine.

### 5.3. The DT (K-M-DW) and DT (T-WM) Multiclassifier Models

As observed in Section 5.2, the dishwasher and Washing Machine, and the microwave and toaster tend to influence each other negatively while the DT model is making decision splits, and, therefore, we hypothesized that these two sets of appliances should not be trained in the same model. Next, we prove this hypothesis by exploring whether the predicted outcome will improve for these four appliances if the models do not attempt to distinguish these appliances during the splitting process. Here, we compare the PD and ICE plots of the DT (K-M-DW) and DT (T-WM) models with those of the DT (K-M-T-WM-DW) model to investigate the impact of the removal of one of these appliances on predicted outcomes of the resulting model.

Comparing Figure 8 and Figure 9 for the three-class and two-class classifiers with the five-class classifier from Figure 4, we can observe that with the removal of the washing Machine or dishwasher and microwave or toaster, EDGE_P has a much stronger influence on the predicted outcome of the microwave, toaster, and washing machine as shown by their PD scores of 1 in Figure 8b and Figure 9a,b. We can also observe in Figure 3 that EDGE_P is the most important feature in both the three-class and two-class classifiers.

In the five-class classifier, we observe EDGE_N has a slightly higher influence on the Toaster prediction than it does on the other features. However, this is only the case if the microwave and toaster are being distinguished using the same classifier as in the earlier five-class classifier. We observe from Figure 3, Figure 10 and Figure 11 that EDGE_N has very little influence on any of the predicted outcomes.

Even though there is a slight rise in the PD plot of the kettle with DURATION as a feature, as shown in Figure 12a, there are many instances of confusion. Intuitively, the kettle, toaster, and microwave have a shorter operating duration, hence DURATION as a feature is not of much significance for these three appliances as also observed in the low PD scores in Figure 12b and Figure 13a. This can also be seen in Figure 7c, which shows many outliers that can affect the performance of these three appliances. The importance of DURATION remains mostly unchanged for both the dishwasher prediction and the washing Machine prediction as seen by the PD/ICE plots of Figure 12c and Figure 13b. As previously discussed, in the five-class classifier, the kettle is likely to be confused with either the washing machine or dishwasher. Therefore, to avoid this confusion, the model strongly considers DURATION for the dishwasher prediction and EDGE_P for the kettle prediction in a three-class classifier model. The washing machine and toaster have very distinct power signatures and duration and thus can easily be distinguished in a two-class classifier model; hence, the two-class classifier model leans towards EDGE_P as a more influential feature for distinguishing the toaster and washing machine. Moreover, we can observe in Figure 3 that EDGE_P is the only important feature in the 2-class classifier. Even the individual instances of EDGE_P are well clustered in Figure 9.

In summary, the kettle, toaster, washing machine, and microwave are expected to work best with EDGE_P and dishwasher with DURATION as features. To confirm this, we performed testing on an unseen portion of the same House 2 dataset with three-class and two-class classifier models embedding feature selection. Table 5 and Table 6 show the resulting F-scores when the models were first built with all three features, and second, with EDGE_P and DURATION as features selected for the three-class classifier and EDGE_P as the selected feature for the two-class classifier.

As hypothesised, comparing Table 3 with Table 5 and Table 6, we can see that the microwave, dishwasher, and washing machine performance has increased when separating the washing machine from the dishwasher during classification and when separating the microwave and the toaster. With the removal of the toaster, the microwave performance increases in F-score from 88% (in a five-class classifier, Table 3) to 91% (in a three-class classifier, Table 5). This is not the case with the toaster because toaster activation in the testing set and unknown appliances “Other” are affecting its performance. Since it has the lowest power signature among the five targeted appliances, unknown appliances “Other” are causing false positives as can be observed in the confusion matrix of Table 7.

We can observe no change in the performances in Table 5 and Table 6, which show the results when the models were trained without and with feature selection, nor can this be observed in their corresponding confusion matrices in Table 7 and Table 8. This is because in the three-class and two-class classifiers, the predicted outcomes are not as influenced by EDGE_N as they are by the EDGE_P and DURATION features, as observed by their PD plots.

## 6. Evaluating Generalisability and Transferability in Relation to the Predicted Outcomes

In this section, we explore generalisability across the same REFIT dataset and transferability across the UK-DALE dataset first by comparing the PD and ICE plots between the predicted outcomes discussed in Section 5.3 and the actual outcomes observed on the testing set. Since the two classifiers (three-class and two-class) were shown to outperform the five-class classifier, we explore generalisability for the DT (K-M-DW) and the DT (T-WM) classifiers.

### 6.1. Generalisability across the UK REFIT Dataset

To show generalisability, the models were trained using REFIT House 2 and tested using unseen REFIT House 6 data using only the important features. From the trained model described in Section 5.3, it was observed that the three-class classifier model considers EDGE_P strongly for prediction “Kettle” and “Microwave” and DURATION for the “Dishwasher” prediction. The two-class classifier model considers EDGE_P strongly for prediction “Toaster” and “Washing Machine”.

Comparing the PD and ICE plots of the EDGE_P of the kettle in Figure 14a with those in Figure 8a, we can observe that both plots have a similar shape and value with the predicted score, showing a sharp rise from 0 to about 0.66-0.96 after about 1800 watts. We expect the kettle to retain its good performance as when tested in House 2. A similar observation can be made comparing the PD and ICE plots of the DURATION feature of the dishwasher in Figure 12c and Figure 15c. However, we observe a drop in the prediction plot from 0.66 to 0.54 around 390 s which is due to the inconsistency in operating the duration between the edges of the dishwasher in Houses 2 and 6. This can be seen in Figure 12c where the DURATION instances are well clustered about DURATION instances between 570 s and 779 s unlike in Figure 15c, where DURATION instances are very few and scattered from 390 s to 1047 s. With this drop, we should expect the classification performance metrics of the dishwasher to drop a bit when tested in House 6. This is evident when we compare Table 5 and Table 9. Furthermore, the drop in the microwave F-score performance is only due to the influence of the unknown appliances including those of the washing machine and toaster as shown by the drop in the precision score. As predicted by the PD and ICE plots, the microwave is not confused with the kettle or dishwasher, as shown by the recall score of 1 for all appliances and in the confusion matrix of Table 10.

Comparing PD and ICE plots of EDGE_P for the Toaster and washing machine in Figure 16 with those in Figure 9, we can observe that both plots have similar a shape and value. Therefore, we expect the classification performance metrics of the toaster and washing Machine to not change much when tested in House 6. Indeed, as seen in Table 10b, the toaster and washing machine are not confused with each other, resulting in a recall of 1 as in Table 11, but Precision drops due to false positives from “Other” appliances, which include unknown appliances.

### 6.2. Transferability to UK-DALE Dataset

In this experiment, the models were trained using House 2 of the UK REFIT dataset and tested on unseen House 1 of the UK-DALE dataset.

Comparing the PD and ICE plots of the EDGE_P of the kettle in Figure 17a with those in Figure 8a, we can observe that both plots have a similar shape and value, with the predicted score of showing a sharp rise from 0 to about 0.66–0.81 after about 1800 watts. We can also observe a sharp rise of the PD plot for the dishwasher in from 0.66 in Figure 18c to 0.80 in Figure 12c. Hence, we expect the kettle and dishwasher to retain their good performance as when tested in House 2. This is indeed the case when we compare Table 5 and Table 12.

The unknown “Others” appliance class affects the microwave performance, as seen in Table 13a. Among the three appliances, the microwave has the lowest power signature and is therefore most likely to be hidden by the unknown appliances in“Others”.

Comparing the PD and ICE plots of EDGE_P for the toaster and washing machine in Figure 19 with those in Figure 9, we can observe that both plots have a similar shape and value. Therefore, we expect the recall performance of the toaster and washing machine to be high; this is indeed the case, with a recall score of 1, as shown in Table 14. The precision score of less than 1 is due to the false positives from few activations from appliances other than the toaster and washing machine.

### 6.3. Benchmarking

Finally, in Table 15, we compared the performance of our proposed classifiers trained on REFIT House 2 and tested on both REFIT House 2 (as presented in Section 5 following feature selection) and REFIT House 6 (as presented in Section 6.2), with the reviewed, event-based, low-rate NILM multiclass DT implementation of [14] and the event-based semisupervised (SGSP) and unsupervised (UGSP) graph signal processing implementations with a dynamic time warping (DTW) distance measure of [32], where a one-against-all approach for each class, one at a time, was adopted for multiclass classification. We also benchmarked against the best performing DT results of [20] for the dishwasher and washing machine in REFIT House 2 with median filtering as the preprocessing and a postprocessing steps. Benchmarks were trained and tested on REFIT Houses 2 and 6, and both require small amounts of data for training, similarly to the proposed DT multiclass classifiers.

We observe that the proposed DT-based 5-classifier approach offers the best performance for the Toaster and Kettle, although the Kettle generally performs well for all DT-based multiclass classifiers. It is worth noting the DT implementation DT H2 [14] classified Fridge, Kettle, Toaster, Microwave and Dishwasher. While it also classified one event to 5 appliances, it did not include the Washing Machine, which is why the Dishwasher has better performance than the equivalent proposed DT (K-M-T-DW-WM) classifier. The proposed DT (K-M-T-DW-WM) classifier outperforms that of DT H2 [14] for Microwave and Toaster, due to the consideration of all 3 features in the proposed approach unlike the average power value considered as feature in all the benchmarks. As observed in Section 5, explainability-informed class composition for each classifier and feature selection does improve classification performance of all appliances except toaster, as shown by F-scores indicated in bold for the DT (K-M-DW) and DT (T-MW) models. Toaster suffers from the influence of other unknown appliances, which have similar EDGE_P and EDGE_N features. Furthermore, including dishwasher and washing machine in the same classifier causes confusion of the two appliances and resulting poor performance for both appliances as evidenced by DT H2 [20] results, which are in line with our DT (K-M-T-WM-DW) classifier results. Comparing the performance of REFIT House 2 trained DT multiclassifiers on REFIT House 6 with those of the SGSP and UGSP classifiers tested on House 6, we can see that the proposed DT multiclassifiers achieve better performance for Kettle, Dishwasher, Microwave and Toaster.

## 7. Conclusions

In this paper, we propose and validate a methodology for explainability-informed appliance-level feature selection and multiclass classifier design by (i) first predicting the performance of a trained multiclass classifier, then validating the prediction via testing; (ii) quantifying the relative feature importance for each appliance within the multiclass classifier, which in turn informs class composition of the multiclass classifier models with explainability-informed feature selection; and (iii) predicting why a particular trained classifier model will generalise well for individual appliances across other houses in the same dataset or transfer well to other houses in different datasets.

The explainability tools, namely PD and ICE plots, help a human visualise how features affect the model in general as well as each predicted outcome. They also help algorithm developers to understand which appliances can be confused with each other through visualisation of the distinguishing features of each appliance. For example, the kettle, toaster, and washing machine are expected to be detected more accurately with EDGE_P as the distinguishing feature, and the dishwasher with DURATION as the feature. To distinguish between the toaster and microwave, EDGE_N is needed. This in turn informs feature selection per appliance and multiclass classifiers that avoid misclassification and improve the classification performance of each appliance. This results in significant improvement in the classification accuracy of 36% for the washing machine and 22% for the dishwasher, maintaining the performance of the kettle and microwave.

When exploring the generalisability of the model trained on House 2 of the REFIT dataset to House 6 in the same REFIT dataset, we expect from the PD and ICE plots for kettle and microwave to retain their performance but dishwasher to drop a little due to differences in DURATION, as confirmed by the classification score during testing. While the PD and ICE plots show that the toaster and washing machine are well distinguished and will retain their performance, as evidenced by a recall of 1, we also observe a drop in F-score due to false positives from unknown appliances. Finally, transferability to another dataset can also be estimated by looking at the actual outcomes on a sample of the testing set of the target house before deployment, thus saving resources on testing effort. Specifically, the PD plots show that the kettle and dishwasher would transfer well from the REFIT to the UK-DALE dataset, and similarly for the toaster and washing machine, and these are confirmed by the F-score and confusion matrices, with similar performance to the REFIT house 2.

Even though the proposed explainability-informed feature selection and class composition of multiclassifier models has the ability to transfer across datasets with minimum performance drop, resulting in the appliances within the multiclassifier model not being confused with each other, it cannot avoid the influence of unknown/unlabelled appliances, which have similar features. Future work will investigate leveraging unsupervised methods to help identify these unknown appliances, as well as to generalise the proposed explainability-informed feature selection and class composition for multiclassifier models for a larger range of appliances such as electric vehicles, air conditioners, and heat pumps.

## Figures and Tables

**Figure 1 sensors-23-04845-f001:**
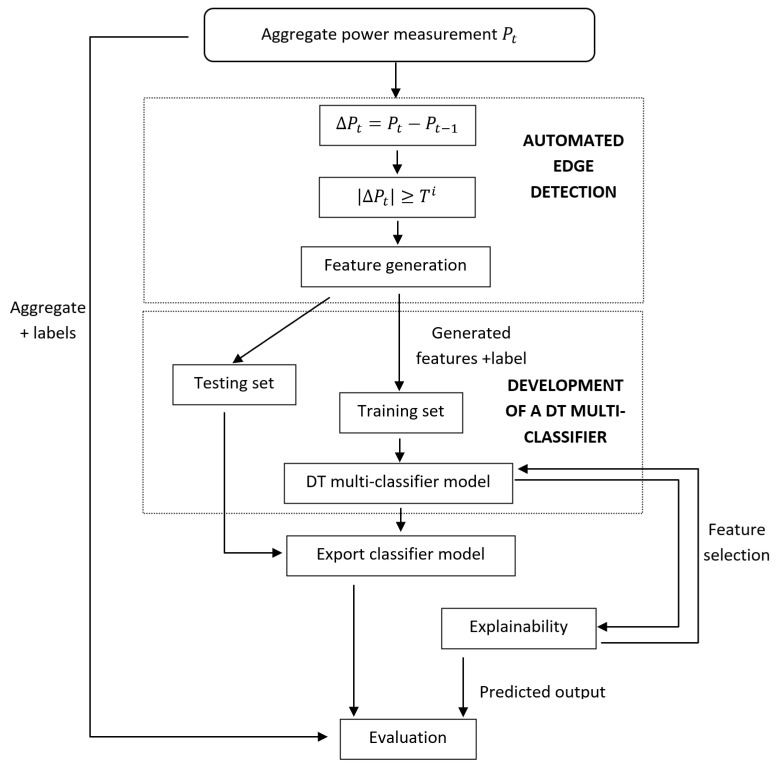
Block Diagram of Methodology.

**Figure 2 sensors-23-04845-f002:**
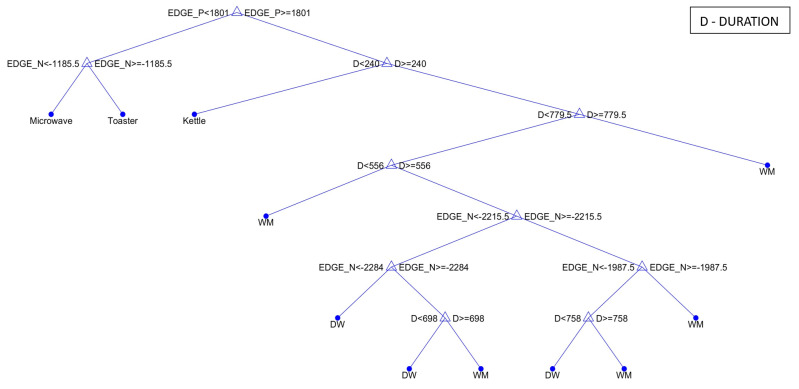
Decision Tree for a 5-Class Classifier.

**Figure 3 sensors-23-04845-f003:**
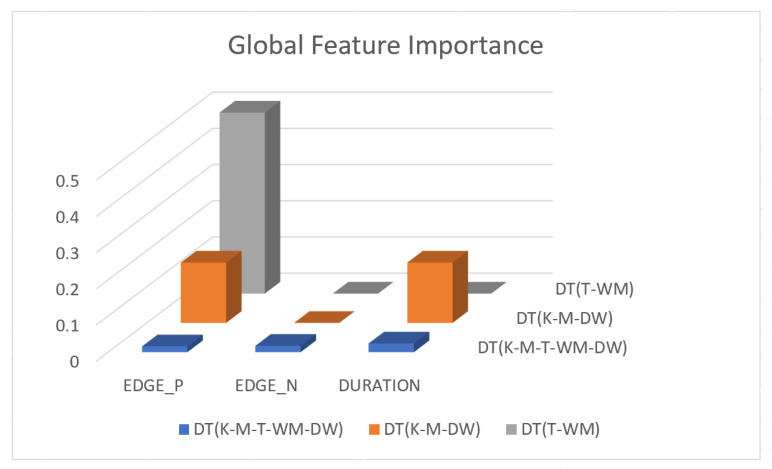
Global feature Importance for DT (K-M-T-WM-DW), DT (K-M-DW), and DT (T-WM) models.

**Figure 4 sensors-23-04845-f004:**
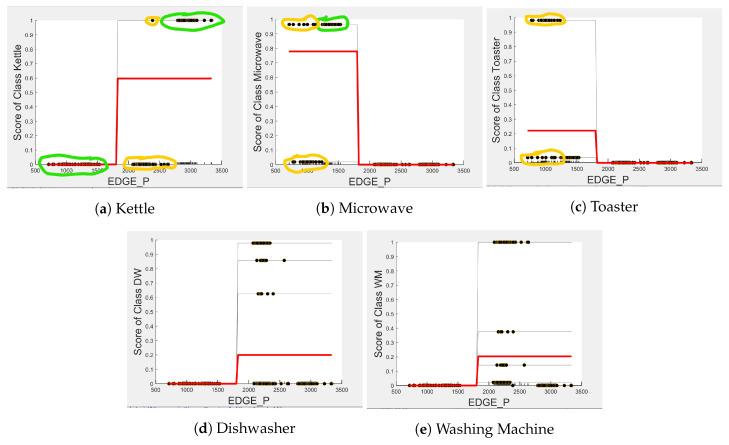
PD and ICE Plots for Predicted Outcome vs. EDGE_P for DT (K-M-T-WM-DW).

**Figure 5 sensors-23-04845-f005:**
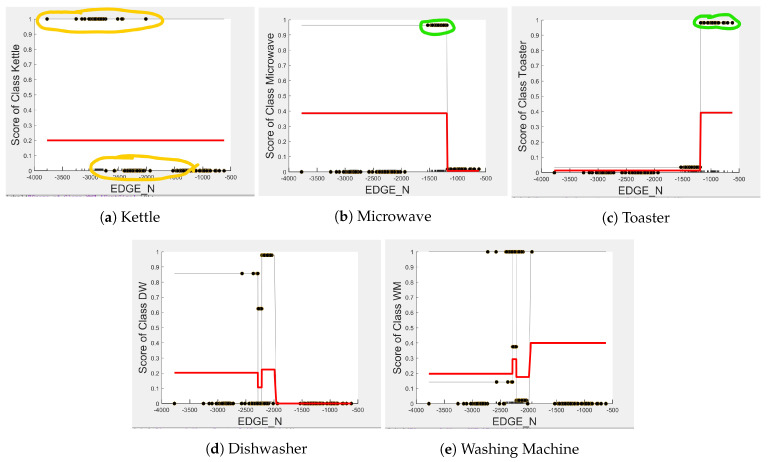
PD and ICE Plots for Predicted Outcome vs. EDGE_N for DT (K-M-T-WM-DW).

**Figure 6 sensors-23-04845-f006:**
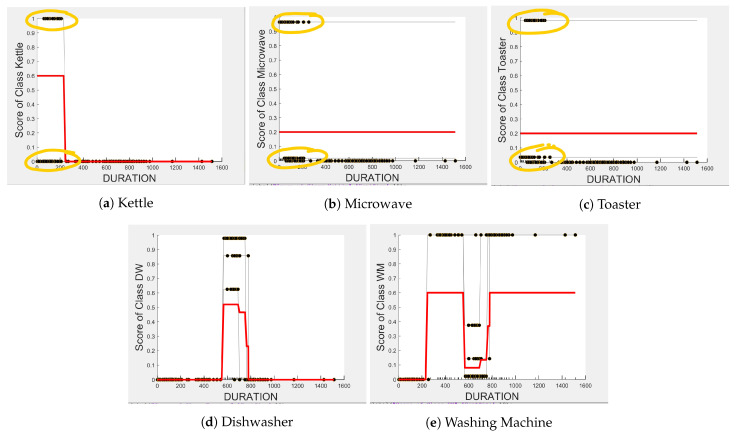
PD and ICE Plots for Predicted Outcome vs. DURATION for DT (K-M-T-WM-DW).

**Figure 7 sensors-23-04845-f007:**
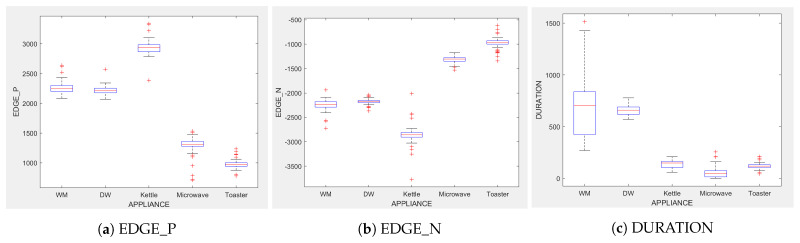
Box plots showing the data distribution of each feature for each appliance.

**Figure 8 sensors-23-04845-f008:**
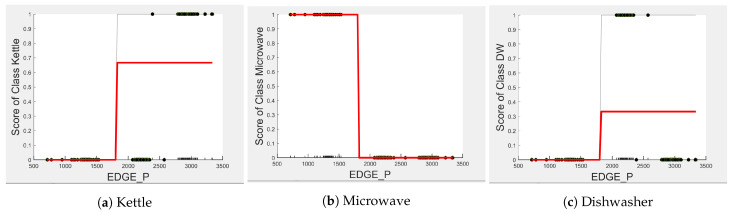
PD and ICE Plots for Predicted Outcome vs. EDGE_P for the DT (K-M-DW) Model.

**Figure 9 sensors-23-04845-f009:**
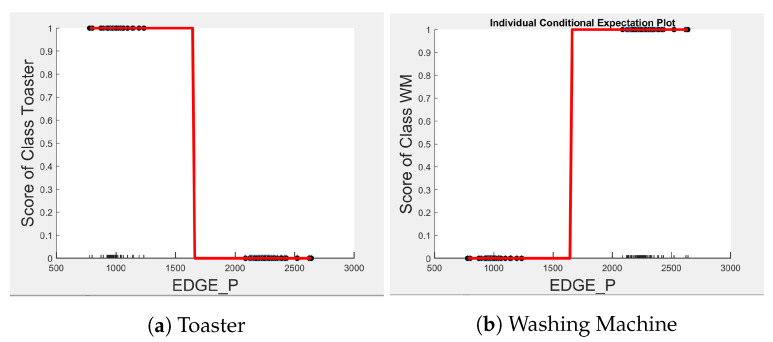
PD and ICE Plots for the Predicted Outcome vs. EDGE_P for the DT (T-WM) Model.

**Figure 10 sensors-23-04845-f010:**
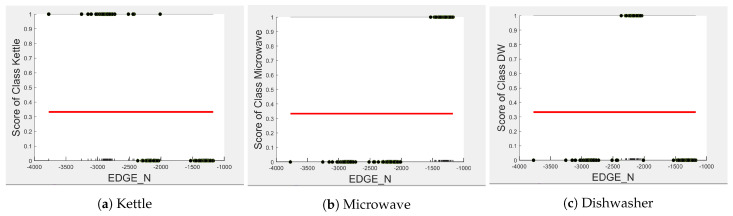
PD and ICE Plots for the Predicted Outcome vs. EDGE_N for the DT (K-M-DW) Model.

**Figure 11 sensors-23-04845-f011:**
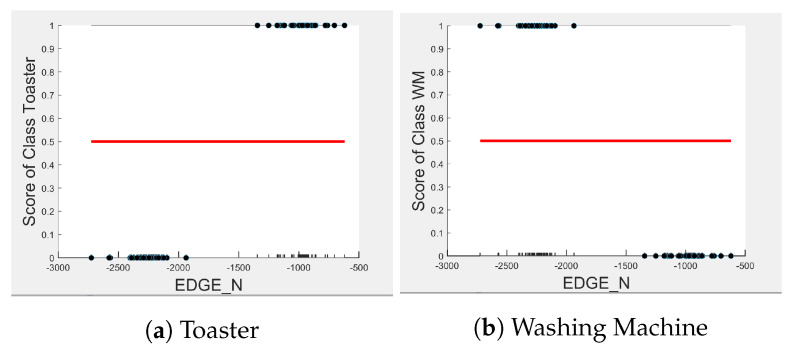
PD and ICE Plots for the Predicted Outcome vs. EDGE_N for the DT (T-WM) Model.

**Figure 12 sensors-23-04845-f012:**
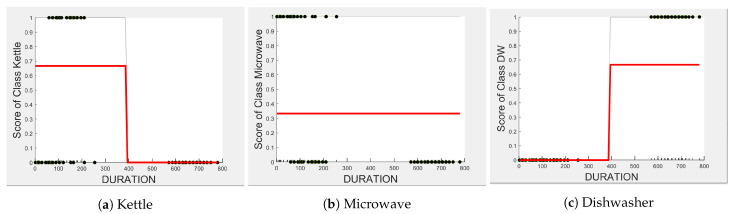
PD and ICE Plots for the Predicted Outcome vs. DURATION for the DT (K-M-DW) Model.

**Figure 13 sensors-23-04845-f013:**
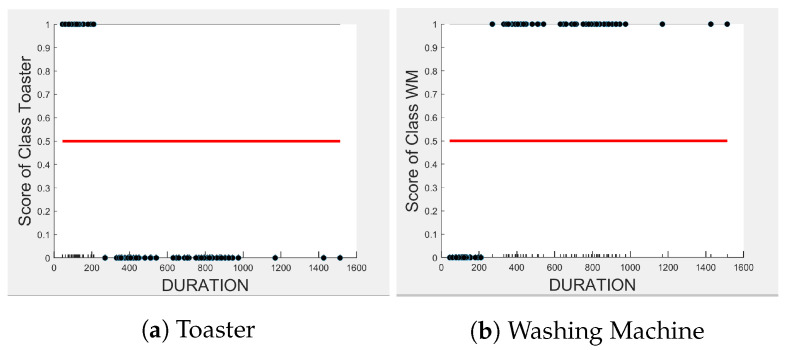
PD and ICE Plots for the Predicted Outcome vs. DURATION for the DT (T-WM) Model.

**Figure 14 sensors-23-04845-f014:**
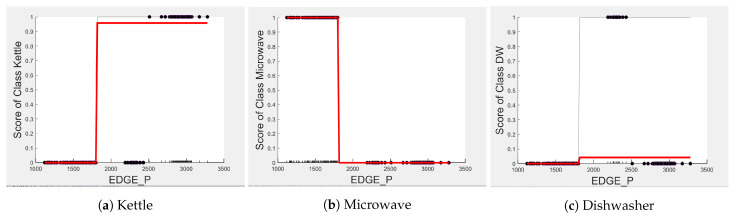
PD and ICE Plots for the Actual Outcome on the Testing set vs. EDGE_P for the DT (K-M-DW) Classifier.

**Figure 15 sensors-23-04845-f015:**
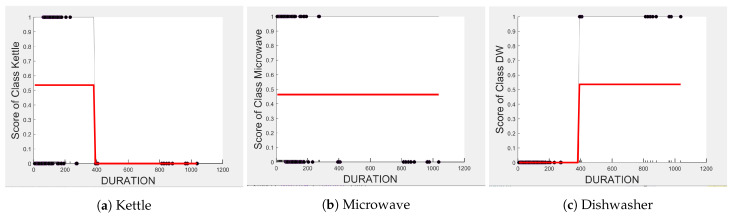
PD and ICE Plots for the Actual Outcome on the H6 testing set vs. DURATION for the DT (K-M-DW) Model.

**Figure 16 sensors-23-04845-f016:**
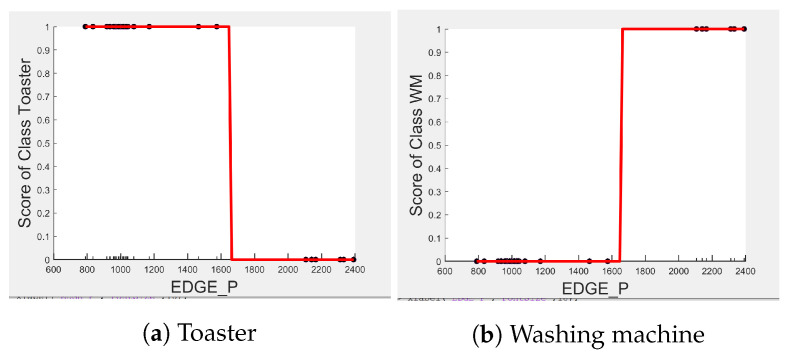
PD and ICE Plots for the Actual Outcome on H6 testing set vs. EDGE_P for the DT (T-WM) Model.

**Figure 17 sensors-23-04845-f017:**
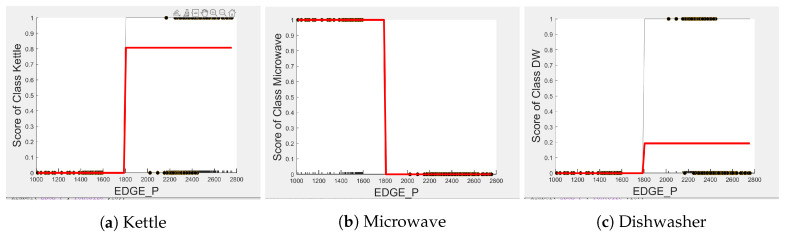
PD and ICE Plots for the Actual Outcome on the UK-DALE Testing Set vs. EDGE_P for the DT (K-M-DW) Classifier.

**Figure 18 sensors-23-04845-f018:**
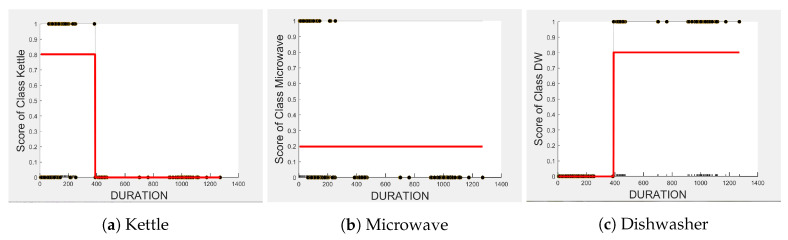
PD and ICE Plots for the Actual Outcome on the UK-DALE Testing Set vs. DURATION for the DT (K-M-DW) Model.

**Figure 19 sensors-23-04845-f019:**
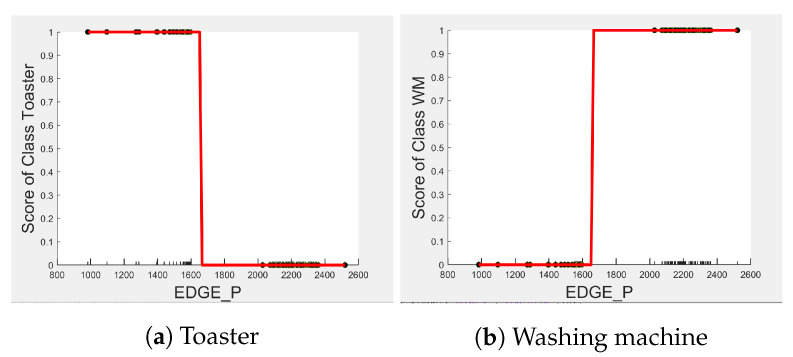
PD and ICE Plots for the Actual Outcome on the UK-DALE Testing Set vs. EDGE_P for the DT (T-WM) Model.

**Table 1 sensors-23-04845-t001:** Edge Detection Performance.

Dataset	Number of Ground-Truth Events	% Events Detected	% Events Due to Unknown Appliances	${\mathrm{NM}}^{\left(T\right)}$
REFIT H2	568	92%	9%	77%
REFIT H6	407	90%	15%	84%
UK-DALE H1	509	96%	6%	79%

**Table 2 sensors-23-04845-t002:** Execution Time.

DT Model	No. Features	REFIT H2		REFIT H2		REFIT H6		UK-DALE H1	
		Training (2 Months)	No. Samples	Testing (3 Months)	No. Samples	Testing (1 Month)	No. Samples	Testing (4 Months)	No. Samples
DT (K-M-T-WM-DW)	3	39.15 s	275	3.17 s	577				
DT (K-M-DW)	2	20.37 s	165	3.29 s	507	3.41 s	394	3.95 s	429
DT (T-WM)	1	25.91 s	110	3.22 s	126	3.32 s	100	3.67 s	132

**Table 3 sensors-23-04845-t003:** Five Classifier, DT (K-M-T-WM-DW), and F-Score When Different Combinations of the Features Are Used.

Appliance	EDGE_P & EDGE_N	EDGE_P	EDGE_N
	& DURATION	& DURATION	& DURATION
Dishwasher	0.70	0.72	0.70
Washing Machine	0.54	0.56	0.53
Kettle	0.99	0.99	0.99
Microwave	0.88	0.90	0.87
Toaster	0.65	0.80	0.64

**Table 4 sensors-23-04845-t004:** Prediction Confusion Matrix for the DT (K-M-T-WM-DW) Model.

(a) Three Features							
		**Predicted Class**					
		DW	K	M	T	WM	Other
**True Class**	DW	47	0	0	0	29	0
	K	0	185	0	0	0	0
	M	0	0	168	22	0	0
	T	0	0	1	35	0	0
	WM	7	0	0	0	27	0
	Other	4	3	24	15	10	0
(**b**) **EDGE_P & DURATION Features**							
		**Predicted Class**					
		DW	K	M	T	WM	Other
**True Class**	DW	48	0	0	0	28	0
	K	0	185	0	0	0	0
	M	0	0	184	6	0	0
	T	0	0	3	33	0	0
	WM	6	0	0	0	28	0
	Other	4	3	31	8	10	0
(**c**) **EDGE_N & DURATION Features**							
		**Predicted Class**					
		DW	K	M	T	WM	Other
**True Class**	DW	47	0	1	0	28	0
	K	0	182	3	0	0	0
	M	0	0	168	22	0	0
	T	0	0	1	35	0	0
	WM	7	0	1	0	26	0
	Other	4	2	24	16	10	0

**Table 5 sensors-23-04845-t005:** F-scores of the DT (K-M-DW) Classifier on the Unseen Data of REFIT House 2.

	Three Features			EDGE_P & DURATION		
**Appliance**	**PR**	**RE**	**F-Score**	**PR**	**RE**	**F-Score**
Dishwasher	0.88	1	0.94	0.88	1	0.94
Kettle	0.96	1	0.98	0.96	1	0.98
Microwave	0.83	1	0.91	0.83	1	0.91

**Table 6 sensors-23-04845-t006:** F-score of the DT (T-WM) Classifier on the Unseen Data of REFIT House 2.

	Three Features			EDGE_P		
**Appliance**	**PR**	**RE**	**F-Score**	**PR**	**RE**	**F-Score**
Washing Machine	0.67	1	0.80	0.67	1	0.80
Toaster	0.48	1	0.65	0.48	1	0.65

**Table 7 sensors-23-04845-t007:** Prediction Confusion Matrix for the DT (T-WM) Model.

(a) Three Features				
		**Predicted Class**		
		T	WM	Other
	T	36	0	0
**True** **Class**	WM	0	34	0
	Other	39	17	0
(**b**) **EDGE_P Feature**				
		**Predicted Class**		
		T	WM	Other
	T	36	0	0
**True** **Class**	WM	0	34	0
	Other	39	17	0

**Table 8 sensors-23-04845-t008:** Prediction Confusion Matrix for the DT (K-M-DW) Model.

(a) Three Features					
		**Predicted Class**			
		DW	K	M	Other
**True Class**	DW	76	0	0	0
	K	0	185	0	0
	M	0	0	190	0
	Other	10	7	39	0
(**b**) **EDGE_P & DURATION Features**					
		**Predicted Class**			
		DW	K	M	Other
**True Class**	DW	76	0	0	0
	K	0	185	0	0
	M	0	0	190	0
	Other	10	7	39	0

**Table 9 sensors-23-04845-t009:** Actual Performance of DT (K-M-DW) classifier on unseen REFIT House 6.

Appliance	PR	RE	F-Score
Dishwasher	0.70	1	0.82
Kettle	0.99	1	0.99
Microwave	0.74	1	0.85

**Table 10 sensors-23-04845-t010:** Confusion Matrix of the DT (K-M-DW) and DT (T-WM) Model Tested on REFIT House 6.

**(a) EDGE_P & DURATION Features**					
	**Predicted Class**				
		DW	K	M	Other
**True Class**	DW	14	0	0	0
	K	0	164	0	0
	M	0	0	154	0
	Other	6	1	55	0
**(b) EDGE_P Feature**					
		**Predicted Class**			
			T	WM	Other
**True Class**		T	32	0	0
		WM	0	6	0
		Other	50	12	0

**Table 11 sensors-23-04845-t011:** Actual Performance of the DT (T-WM) Classifier on unseen REFIT House 6.

Appliance	PR	RE	F-Score
Washing Machine	0.33	1	0.50
Toaster	0.39	1	0.56

**Table 12 sensors-23-04845-t012:** Actual Performance of the DT (K-M-DW) Classifier on Unseen UK-DALE House 1.

Appliance	PR	RE	F-Score
Dishwasher	0.90	1	0.95
Kettle	0.98	1	0.99
Microwave	0.76	1	0.87

**Table 13 sensors-23-04845-t013:** Confusion Matrix of the DT (K-M-DW) and DT (T-WM) Model Tested on UK-DALE House 1.

**(a) EDGE_P & DURATION Features**					
	**Predicted Class**				
		DW	K	M	Other
**True Class**	DW	76	1	0	0
	K	0	239	0	0
	M	0	0	78	0
	Other	8	3	24	0
**(b) EDGE_P Feature**					
		**Predicted Class**			
			T	WM	Other
**True** **Class**		T	32	0	0
		WM	0	65	0
		Other	20	15	0

**Table 14 sensors-23-04845-t014:** Actual Performance of the DT (T-WM) Classifier on Unseen UK-DALE House 1.

Appliance	PR	RE	F-Score
Washing Machine	0.81	1	0.90
Toaster	0.62	1	0.76

**Table 15 sensors-23-04845-t015:** F-SCORE Performance Comparison of the Proposed DT-Based Multiclassifiers and Other Event-based Multiclass Classifiers on the REFIT dataset.

Appliance	K	WM	DW	MW	T
DT (K-M-T-WM-DW)H2	**0.99**	0.56	0.72	0.90	**0.80**
DT H2 [14]	0.9		0.80	0.83	0.70
DT (K-M-DW)H2	0.98		**0.94**	**0.91**	
DT (T-WM)H2		**0.80**			0.65
DT H2 [20]		0.52	0.77		
SGSP H2 [32]	0.87	0.64	0.63	0.68	0.58
UGSP H2 [32]	0.90	0.70	0.61	0.79	0.72
DT (K-M-DW)H6	**0.99**		0.82	0.85	
DT (T-WM)H6		0.50			0.56
SGSP H6 [32]	0.79		0.57	0.63	0.45
UGSP H6 [32]	0.77		0.69	0.70	0.44

K—kettle; WM—washing machine; DW—dishwasher; M—microwave; T–toaster; H2—REFIT House 2; H6—REFIT House 6.

## Data Availability

Publicly available datasets were analysed in this study. These data can be found at https://doi.org/10.15129/9ab14b0e-19ac-4279-938f-27f643078cec and https://jack-kelly.com/data/, accessed on 17 December 2022.

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
