# Peer review of "Explainability-Informed Feature Selection and Performance Prediction for Nonintrusive Load Monitoring†"

_sensors, 2023, doi:10.3390/s23104845_

Round 1

Reviewer 1 Report

Authors proposed methodology titled as "Explainability-Informed Feature Selection and Performance Prediction for Non-Intrusive Load Monitoring." After reading manuscript, there are certain suggestions which are mentioned below to improve the quality of manuscript.

1. In Title " Explainability-Informed Feature Selection"  is mentioned but in abstract there is no sentence regarding explainability.

2. Abstract need to be modified. Summary in abstract and discussions in various sections are not matching. Further, include the numerical results.

3. What is the specific reason to choose Decision tree as a classifier. There are several ML models which can be used as an Explainable model.

4. Introduction and background section can be combined. It is highly recommended to shorten the literature review. Both section included literatures only.

5. Authors utilized feature selection. Precisely include details in table which features are selected and which are discarded.

6. Box and whsiker plot is a good visualization techniques which can detect outliers.Kindly include box and whisker plot in revised manuscript and show the feature distribution.Refer following paper and add discussion in revised version:

a. https://www.sciencedirect.com/science/article/abs/pii/S0375960121006642

b. https://www.mdpi.com/2409-9279/3/4/64

7. It is always expected to apply 10-fold cross validation and highlights the prediction results.Training and testing gives biased prediction results due to random split of data.To get more idea,refer 6a.

8. Conclusion section should be re-written. Kindly include the results outcomes in detail.

9. Limitations and future scope of methodology should be included.

Minor editing of English language required

Author Response

Dear Reviewer 1,

Attached are the response to your comments and revised manuscript. Thank you.

Regards,

Reviewer 2 Report

The paper is well written and structured, I have only but a few comments:

1. Benchmarking against other papers is great and adds value to the paper. At the same time though I would prefer to see finally more results on different appliances than the tpical ones used always. In any case, EVs, heat pumps, AirCos, solar are the appliances/activities of interest anymore. Could the authors extend their results with more appliances?

2. Although the references list is quite rich there are recent papers on real-time, scalable and light-weight NILM approaches that are not mentioned. Such solutions could run on the edge and exploit NILM potential.

3. A discussion section on computational effort/burden is needed. That will shed light on where exactly such a solution can run, how scalable it is etc 

Author Response

Dear Reviewer 2,

Attached are the response to your comments and a revised manuscript. Thank you.

Regards,

Reviewer 3 Report

  • I have no further comment and agree to publication, congratulations to the authors.

Author Response

Dear Reviewer 3,

We would like to thank you for reviewing our article titled “Explainability-Informed Feature Selection and Performance Prediction for Non-Intrusive Load Monitoring” and providing us with useful suggestions.

Regards

Round 2

Reviewer 1 Report

Authors corrected manuscript as per suggestion and addressed all comments.

Language is fine.

Reviewer 2 Report

no further comments